# Differentiation between Parkinson’s Disease and the Parkinsonian Subtype of Multiple System Atrophy Using the Magnetic Resonance T1w/T2w Ratio in the Middle Cerebellar Peduncle

**DOI:** 10.3390/diagnostics14020201

**Published:** 2024-01-17

**Authors:** Jiaqi Wang, Atsuhiko Sugiyama, Hajime Yokota, Shigeki Hirano, Tatsuya Yamamoto, Yoshitaka Yamanaka, Nobuyuki Araki, Shoichi Ito, Friedemann Paul, Satoshi Kuwabara

**Affiliations:** 1Department of Neurology, Graduate School of Medicine, Chiba University, Chiba 260-8670, Japan; kingplus735517099@gmail.com (J.W.); tatsuya-yamamoto@mbc.nifty.com (T.Y.); sito@faculty.chiba-u.jp (S.I.); kuwabara-s@faculty.chiba-u.jp (S.K.); 2Diagnostic Radiology and Radiation Oncology, Graduate School of Medicine, Chiba University, Chiba 260-8670, Japan; 3Department of Rehabilitation, Division of Occupational Therapy, Chiba Prefectural University of Health Sciences, Chiba 261-0014, Japan; 4Urayasu Rehabilitation Education Center, Chiba University Hospital, Urayasu 279-0023, Japan; 5Department of Medical Education, Graduate School of Medicine, Chiba University, Chiba 260-8670, Japan; 6Experimental and Clinical Research Center, Max Delbrueck Center for Molecular Medicine and Charité-Universitätsmedizin Berlin, 10117 Berlin, Germany; 7NeuroCure Clinical Research Center, Charité-Universitätsmedizin Berlin, 10117 Berlin, Germany; 8Einstein Center for Neurosciences, Charité-Universitätsmedizin Berlin, 10117 Berlin, Germany; 9Department of Neurology, Charité-Universitätsmedizin Berlin, 10117 Berlin, Germany

**Keywords:** Parkinson’s disease, multiple system atrophy, magnetic resonance imaging, middle cerebellar peduncle

## Abstract

Multiple system atrophy with predominant parkinsonism (MSA-P) can hardly be distinguished from Parkinson’s disease (PD) clinically in the early stages. This study investigated whether a standardized T1-weighted/T2-weighted ratio (sT1w/T2w ratio) can effectively detect degenerative changes in the middle cerebellar peduncle (MCP) associated with MSA-P and PD and evaluated its potential to distinguish between these two diseases. We included 35 patients with MSA-P, 32 patients with PD, and 17 controls. T1w and T2w scans were acquired using a 1.5-T MR system. The MCP sT1w/T2w ratio was analyzed via SPM12 using a region-of-interest approach in a normalized space. The diagnostic performance of the MCP sT1w/T2w ratio was compared between the MSA-P, PD, and controls. Patients with MSA-P had significantly lower MCP sT1w/T2w ratios than patients with PD and controls. Furthermore, MCP sT1w/T2w ratios were lower in patients with PD than in the controls. The MCP sT1w/T2w ratio showed excellent or good accuracy for differentiating MSA-P or PD from the control (area under the curve (AUC) = 0.919 and 0.814, respectively) and substantial power for differentiating MSA-P from PD (AUC = 0.724). Therefore, the MCP sT1w/T2w ratio is sensitive in detecting degenerative changes in the MCP associated with MSA-P and PD and is useful in distinguishing MSA-P from PD.

## 1. Introduction

Multiple system atrophy (MSA) is an adult-onset neurodegenerative disorder clinically characterized by a combination of autonomic dysfunction, parkinsonism, cerebellar ataxia, and pyramidal signs [1]. MSA is principally divided into two clinical subtypes: MSA with parkinsonism (MSA-P) and MSA with cerebellar ataxia (MSA-C) [1]. Its prevalence is 3.4–4.9 per 100,000 people [1]. Parkinson’s disease (PD) is the second most common neurodegenerative disease, with a global prevalence of more than 6 million individuals. PD results from the progressive loss of dopaminergic neurons of the substantia nigra projecting to the striatum [2]. PD and MSA-P are both α-synucleinopathies that share parkinsonian symptoms as the predominant motor symptoms [3]. However, patients with MSA-P generally show poorer levodopa response and faster progression than those with PD [4,5,6]. In patients with MSA, the mean survival from the initial clinical presentation is 6–10 years, with few of them surviving for more than 15 years [1]. Moreover, these patients are at risk of sudden death secondary to asphyxia, probably resulting from vocal cord abductor paralysis, abnormal breathing control, sputum and food matter blockage, and cardiac autonomic dysfunction [7]. Therefore, timely and precise differentiation between these two conditions is critical for optimal patient management. Considering the need for a disease-modifying therapy in the future, early and precise diagnosis is important for early intervention and inclusion in clinical trials. However, misdiagnosis rates are high in several studies of autopsy-confirmed MSA, especially in the early disease course, and Lewy body disease and PD are the most common underlying pathologic findings in patients clinically misdiagnosed with MSA [8,9,10].

Diagnostic markers that discriminate between MSA-P and PD are usually identified using conventional magnetic resonance imaging (MRI). Pathologically, although the nigrostriatal system is the primary pathological site, MSA-P-associated degeneration can be widespread and commonly includes the olivopontocerebellar system [11,12]. Within the olivopontocerebellar system, the pontocerebellar fibers in the middle cerebellar peduncle (MCP) are among the earliest affected regions [13]. Therefore, detecting changes associated with MCP degeneration using conventional MRI may be a potential diagnostic approach for differentiating MSA-P from PD. Indeed, MCP width and changes in diffusion-weighted imaging (DWI) and diffusion tensor imaging (DTI) in the MCP have been reported to be useful for differentiating these two conditions [14,15,16,17,18,19,20,21]. Recent studies have shown the utility and unique quantitative contrast provided by the ratio between the signal intensities of T1- and T2-weighted (T1w/T2w ratio) images, considering their high test–retest reliability and sensitivity to neurodegenerative changes [22,23,24]. Standardization of the T1w/T2w ratio (sT1w/T2w ratio) has been proposed, allowing for a meaningful comparison between subjects and scanners by creating scaled intensity while correcting for inhomogeneities in receiver coil sensitivity [25]. We previously reported that using the sT1w/T2w ratio for evaluating MCP can help detect early MSA-C-related degenerative changes (e.g., myelin loss), considering that it shows high diagnostic accuracy in differentiating patients with MSA-C from healthy individuals and those with spinocerebellar ataxia [26,27]. However, the usefulness of MCP sT1w/T2w ratio in differentiating between MSA-P and PD remains unreported.

Therefore, this study aimed to determine the utility of the sT1w/T2w ratio for detecting MSA-P- and PD-related degenerative changes in MCP and assess its potential for distinguishing these two disorders.

## 2. Materials and Methods

### 2.1. Participants

The Institutional Review Board of Chiba University Graduate School of Medicine approved this retrospective study. We utilized clinical and radiological data obtained during routine clinical practice; thus, the need for written informed consent was waived. This study was conducted between October 2022 and December 2022. The inclusion criteria for patients with MSA-P were as follows: 1.5-T MR images acquired between April 2010 and March 2020 and a diagnosis of clinically established or probable MSA-P according to recent diagnostic criteria [28], with the terms “established” and “probable” reflecting the levels of diagnostic certainty. According to the predominant clinical symptomatology during MRI, MSA-P was differentiated from MSA-C. The exclusion criterion was a current or previous history of another neuropsychiatric disease. On the basis of these inclusion and exclusion criteria, 35 patients with MSA-P were included in the study. Additionally, patients with clinically established or probable PD based on the MDS Clinical Diagnostic Criteria [29] who underwent 1.5-T MRI between April 2010 and March 2020 and without a current or previous history of another neuropsychiatric disease were identified from our database. From these patients, 35 patients with PD who were age- and sex-matched to the MSA-P group were selected. Furthermore, according to the following inclusion and exclusion criteria, 17 subjects were used as controls. Patients who were referred to our hospital with complaints of dizziness, headache, or lightheadedness, had no neurological abnormalities, and underwent 1.5-T MR imaging between April 2012 and March 2020 were included as the controls, while those with a current or previous history of another neuropsychiatric disease were excluded. After reviewing the MR images, we further excluded two patients with PD because of poor-quality scans for assessment caused by artifacts, and one because of MRI findings suggesting idiopathic normal-pressure hydrocephalus. Finally, the study cohort included 35 patients with MSA-P, 32 patients with PD, and 17 controls. After reviewing the medical records of all patients, we retrieved data on sex, age at MRI, age at onset, disease duration (from symptom onset to MRI scan), Hoehn and Yahr stage, and levodopa equivalent daily dose (LEDD). In patients with MSA-P, we determined whether cerebellar ataxia was present on neurological examination during the MRI scan, and in those with PD, we identified whether tremor was their initial motor symptom.

### 2.2. MRI Acquisition

All MR images were acquired using a 1.5-T MRI system (GE Signa HDxt) with a 12-channel head coil. The following parameters were used for T1w imaging: 3D-IR-SPGR, sagittal plane, TR, 7 ms; TE, 3 ms; flip angle, 15°; FOV, 240 mm × 240 mm; matrix, 256 × 256; voxel size, 0.86 mm × 0.86 mm × 1.5 mm; number of slices, 248; and number of excitations, 1. For T2w imaging, the following were used: 2D-TSE; TR, 4000 ms; TE, 100 ms; FOV, 220 mm × 220 mm; matrix, 320 × 256; voxel size, 0.43 mm × 0.43 mm × 6 mm; number of slices, 23; interslice gap, 6 mm; and number of excitations, 1.

### 2.3. MRI Preprocessing

Before sT1w/T2w ratio calculation, all T1w and T2w images underwent preprocessing [26]. Specifically, intensity inhomogeneity correction was applied to the T1w and T2w images by using N4BiasFieldCorrection [30]. SPM12 was used to linearly coregister 3D T1w images with 2D axial T2w images. In creating brain masks, the coregistered T1w images were skull-stripped using the Brain Extraction Tool with FSL (version 5.0.11) [31] and finalized with FSLmaths. White matter and gray matter brain masks were generated using the FMRIB Automatic Segmentation Tool (FSL FAST) on the coregistered T1w image [32].

### 2.4. T1w/T2w and Standardized T1w/T2w Ratios

The median intensity values of T1w and T2w images in both white and gray matter masks from each participant were calculated using the FSL stats. We divided the median gray matter intensity value in the T1w images by that in the T2w image to generate a scaling factor, which was then used to calculate the sT1w/T2w ratio. To create a scaled T2w image (sT2), we multiplied the T2w image by the scaling factor. Finally, the sT1w/T2w ratio was calculated using the following equation developed by Misaki et al. [24]:$$s\frac{T1w}{T2w}ratio=\frac{T1w-sT2}{T1w+sT2}$$

A schematic sT1w/T2w ratio map was built according to a previous study [26]. The sT1w/T2w ratio map for each participant was registered in the Montreal Neurological Institute 152 space [33] using Advanced Normalization Tools [34]. A Gaussian Kernel with an 8 mm-full width at half maximum was used to spatially smooth each image. Regions of interest in the MCP were defined bilaterally on normalized sT1w/T2w ratio maps by using a validated probabilistic 3D atlas of the cerebellar white matter structure via SPM in Matlab 2014a [35], as described previously [26]. Parcellation at a 90% probability threshold was used. The Atlas registration accuracy was visually verified using the registration tool of SPM12 in Matlab 2014a. The MCP sT1w/T2w ratio value for each participant was calculated as the mean value of the median sT1w/T2w ratios in the left and right MCP regions. Figure 1 shows representative images of the sT1w/T2w ratio maps in patients with MSA-P and PD and controls.

### 2.5. Statistical Analyses

All statistical analyses, except receiver operating characteristic (ROC) curve analyses, were performed using the Statistical Package for the Social Sciences version 27.0 (SPSS Inc., IBM Corp., Chicago, IL, USA). For the ROC curve analyses, we used the JMP pro version 14.2.0 (SAS Institute). To determine demographic differences between the MSA-P, PD, and control groups, we used the chi-square (χ^2^) test and Kruskal–Wallis one-way analysis of covariance (ANOVA) for sex and age at MRI, respectively. Differences in disease duration, age at onset, and LEDD between the MSA-P and PD groups were analyzed using the Mann–Whitney *U* test. To evaluate the differences in Hoehn and Yahr stages between the MSA-P and PD groups, we employed the chi-square (χ^2^) test. Differences in MCP sT1w/T2w ratios among the three groups were assessed using one-way ANOVA, with age as the covariate. Moreover, we used ROC curve studies to evaluate the ability of MCP sT1w/T2w ratios to differentiate the three groups, and the Youden method to determine the optimal cutoff point [36]. The area under the curve (AUC) values of >0.9, >0.8, and >0.7 were considered excellent, good, and fair, respectively. The relationship between the MCP sT1w/T2w ratio and disease duration in patients with MSA-P and PD was evaluated through Spearman correlation analysis. Using Student’s *t* test, we compared the MCP sT1w/T2w ratios between patients with MSA-P with and without cerebellar ataxia. Spearman correlation was also used for evaluating the relationship between the MCP sT1w/T2w ratios and disease duration in each subgroup of patients with MSA-P with and without cerebellar ataxia detected during MRI. Furthermore, we compared the MCP sT1w/T2w ratios between patients with PD with and without tremor onset. The relationship between MCP sT1w/T2w ratios and disease duration in each subgroup of patients with PD with or without tremor onset was assessed through Spearman correlation analysis. A *p*-value of <0.05 indicated statistical significance.

## 3. Results

Table 1 summarizes the demographic and clinical data of the PD, MSA-P, and control groups.

Disease duration was significantly longer in patients with PD than in those with MSA-P. LEDD was also significantly higher in the PD group than in the MSA-P group. Conversely, the MCP sT1w/T2w ratios were significantly lower in the MSA-P group than in the PD and control groups (0.11 ± 0.06 vs. 0.16 ± 0.03 and 0.19 ± 0.03, respectively; *p* = 0.001 and *p* < 0.001).

In addition, patients with PD had significantly lower MCP sT1w/T2w ratios than the controls (*p* = 0.014; Figure 2).

The AUC for differentiating MSA-P from PD was 0.724 (95% confidence interval (CI), 0.603–0.845). The optimal cutoff point of the MCP sT1w/T2w ratio value, estimated by the Youden index, was 0.137989, with sensitivity and specificity of 0.629 and 0.781, respectively. The AUC for distinguishing MSA-P from the control was 0.919 (95% CI, 0.840–0.999). The optimal cutoff point of the MCP sT1w/T2w ratio value was 0.165714, with sensitivity and specificity of 0.857 and 0.882, respectively. Lastly, the AUC for differentiating PD from the control was 0.814 (95% CI, 0.681–0.948), and the optimal cutoff point of the MCP sT1w/T2w ratio value was 0.1732445, with sensitivity and specificity of 0.750 and 0.824, respectively (Figure 3).

Of the 35 patients with MSA-P, 16 had cerebellar ataxia. The MCP sT1w/T2w ratios were significantly lower in patients with MSA-P and cerebellar ataxia than in those without cerebellar ataxia (0.07 ± 0.06 vs. 0.15 ± 0.03; *p* = 0.003). Disease duration did not significantly correlate with the MCP sT1w/T2w ratio in patients with MSA-P (*r* = −0.141, *p* = 0.418). Likewise, disease duration had no significant correlation with the MCP sT1w/T2w ratio in both subgroups of patients with MSA-P with cerebellar ataxia (*r* = −0.318, *p* = 0.230) and those without cerebellar ataxia (*r* = 0.179, *p* = 0.464) (Figure 4).

Of the 32 patients with PD, 15 had tremor onset. The MCP sT1w/T2w ratios were not significantly different between patients with PD with tremor onset and those without tremor onset (0.15 ± 0.02 vs. 0.16 ± 0.03; *p* = 0.166). Disease duration did not significantly correlate with the MCP sT1w/T2w ratio in patients with PD (*r* = −0.076, *p* = 0.680). It also had no significant correlation with the MCP sT1w/T2w ratio in both subgroups of patients with PD with tremor onset (*r* = −0.301, *p* = 0.276) and those without tremor onset (*r* = 0.196, *p* = 0.450) (Figure 5).

## 4. Discussion

This study showed that the MCP sT1w/T2w ratios were significantly lower in MSA-P than in PD or the control and were significantly lower in PD than in the control. ROC curve analysis revealed that the MCP sT1w/T2w ratio showed excellent or good diagnostic performance in differentiating MSA-P or PD from the control, suggesting its sensitivity in detecting MSA-P- or PD-related degenerative changes in the MCP. Moreover, the substantial discriminatory power of the MCP sT1w/T2w ratio described herein implies that it is useful in differentiating MSA-P and PD.

As mentioned, the sT1w/T2w ratio can sensitively detect MSA-P-related degenerative changes in the MCP. Accordingly, our findings showed that patients with MSA-P had significantly lower MCP sT1w/T2w values than the controls. Moreover, the MCP sT1w/T2w ratios excellently differentiated MSA-P from the control, with high sensitivity and specificity. Our results are consistent with previous pathological findings showing that most of the patients with MSA-P exhibit certain degenerative changes in not only the nigrostriatal but also in the olivopontocerebellar system [11,12]. The pontocerebellar fibers in the MCP are among the earliest regions to display MSA-related degenerative changes in the olivocerebellar system [13]. Consistent with the current study results, our previous studies revealed that the MCP sT1w/T2w ratio can differentiate patients with early MSA-C from those with hereditary spinocerebellar ataxia [26,27]. Pathologically, reactive gliosis and the loss of axons and myelin are the MSA-related degenerative changes in the MCP [37], presumably reflected in the decreased MSA sT1w/T2w ratio [27].

In the present study, the MCP sT1w/T2w ratio was lower in patients with MSA-P with cerebellar ataxia than in those without cerebellar ataxia, suggesting its association with cerebellar ataxia in patients with MSA-P. In the correlation analysis, the y-intercept of the regression line for patients with MSA-P with cerebellar ataxia was approximately 0.10, and some of those with MSA-P had low MCP sT1w/T2w ratios despite having a short disease duration. Results showed no correlation between the MCP sT1w/T2w ratio and disease duration in patients with MSA-P with cerebellar ataxia. Therefore, the MCP sT1w/T2w ratio may reflect the degree of damage to the olivopontocerebellar system regardless of disease duration. This result is consistent with a previous MSA-C study finding, that is, the MCP sT1w/T2w ratio correlated with cerebellar ataxia severity and not with disease duration [26]. Motor slowness resulting from parkinsonism makes the assessment of dysmetria or decomposition associated with cerebellar ataxia difficult. Indeed, parkinsonism contributes to miscalculating the ratings of cerebellar ataxia scales, such as the International Cooperative Ataxia Rating Scale (ICARS), in patients with MSA [38]. Therefore, the MCP sT1w/T2w ratio may serve as an objective marker of olivopontocerebellar system damage that cannot be accurately assessed on neurological examinations in patients with MSA-P. Unfortunately, considering the retrospective nature of the present study, we could not assess the association between the MCP sT1w/T2w ratio and validated cerebellar ataxia scales, such as the ICARS and the Scale for the Assessment and Rating of Ataxia.

Our study also suggests that the MCP sT1w/T2w ratio can detect PD-related changes in the MCP. In fact, our patients with PD had significantly lower MCP sT1w/T2w ratios than the controls. Although the pathological hallmark of PD is progressive dopamine neuronal loss within the substantia nigra, increasing evidence suggests that the cerebellum is involved in PD pathophysiology through both pathological changes and compensatory effects [39]. A study using MRI to measure MCP width has shown an inverse correlation between MCP width and disease duration in patients with PD without tremor onset [40]. Studies using DTI or DWI conductometry have shown microstructural changes in the MCP among patients with PD who exhibit freezing of gait or rapid-eye-movement sleep behavior disorders (RBD), respectively [41,42,43]. Unfortunately, given that we could not compare the MCP sT1w/Tw ratios with the patient’s histopathological findings, we cannot confirm the pathological background for the decreased MCP sT1w/T2w ratios in the MCP of patients with PD. However, consistent with our study, another study reported T1w intensity decrement in the MCP of patients with PD [44]. Contrary to the findings of Sako et al. [40], disease duration did not significantly correlate with the MCP sT1w/T2w ratios in patients with PD with or without tremor onset. This inconsistency may have resulted from the differences in patient background. Patients with PD included in this study were older during the MRI and had a longer and more widely distributed disease duration than those included in the previous study. Unfortunately, given the retrospective nature of the current study, we could not assess the association between the MCP sT1w/T2w ratio and other symptoms, such as freezing of gait and RBD. Further studies are required to clarify the association between the MCP sT1w/T2w ratio and the clinical symptoms of PD.

Although patients with PD had a significantly longer disease duration than those with MSA, the MCP sT1w/T2w ratio showed substantial power in differentiating MSA-P and PD in this study. Its usefulness in differentiating MSA-P from PD observed in the current study could be attributed to the differences in the degree of MCP degeneration between the two diseases. Concurrent with our results, MCP width, DTI measurements, and apparent diffusion coefficient measurements have been reported to be useful for differentiating MSA-P from PD [14,15,16,17,18,19,20,21]. Further research is needed to compare the diagnostic performance of these quantitative methods and clarify which approaches are better for detecting early MSA- or PD-related changes in the MCP.

Our study has some limitations that are worth noting. First, the patients with MSA-P and PD were clinically diagnosed without postmortem verification; hence, we cannot exclude the possibility that some had been misdiagnosed. Second, our sample size was relatively small, suggesting the need for further studies to validate the presented findings. Third, considering the retrospective nature of this study, we could not evaluate the relationship between the MCP sT1w/T2w ratio and clinical scales for motor and nonmotor symptoms, such as the Unified Parkinson’s Disease Rating Scale or the Scale for Outcomes in Parkinson’s Disease for Autonomic Symptoms. Finally, given that disease duration was not considered when selecting patients with PD matched for age and sex with the MSA-P group, the patients with PD had a large range of disease duration and longer disease duration than those with MSA-P. Therefore, we cannot rule out the idea that differences in disease duration and a large range of disease duration may have affected our study results. However, disease duration showed no significant correlation with the MCP sT1w/T2w ratio in both the PD and MSA-P groups in this study.

## 5. Conclusions

The MCP sT1w/T2w ratio can sensitively detect MSA- and PD-related degenerative changes in the MCP, indicating its potential utility for differentiating MSA-P from PD.

## Figures and Tables

**Figure 1 diagnostics-14-00201-f001:**
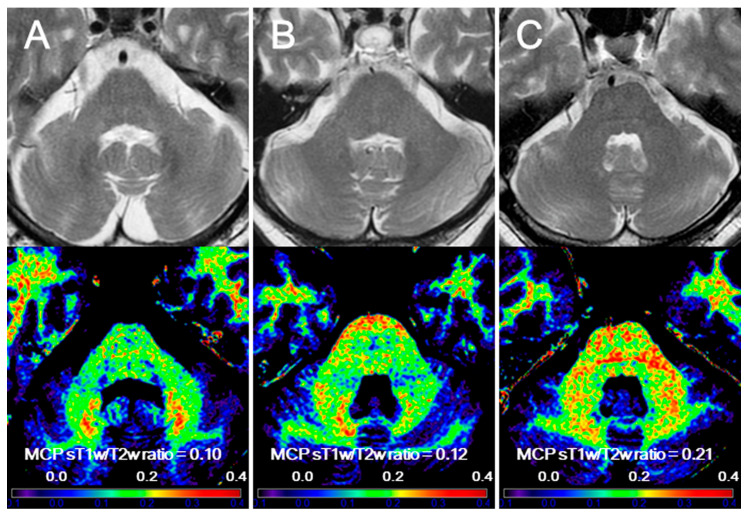
Representative images of standardized T1-weighted/T2-weighted ratio maps in multiple system atrophy with parkinsonism (MSA-P), Parkinson’s disease (PD), and control. (**A**) A 62-year-old man with MSA-P for 4.8 years; (**B**) a 70-year-old woman with PD for 17.8 years; (**C**) a 63-year-old man as the control. Upper row: axial T2-weighted images. Lower row: standardized T1-weighted/T2-weighted ratio maps.

**Figure 2 diagnostics-14-00201-f002:**
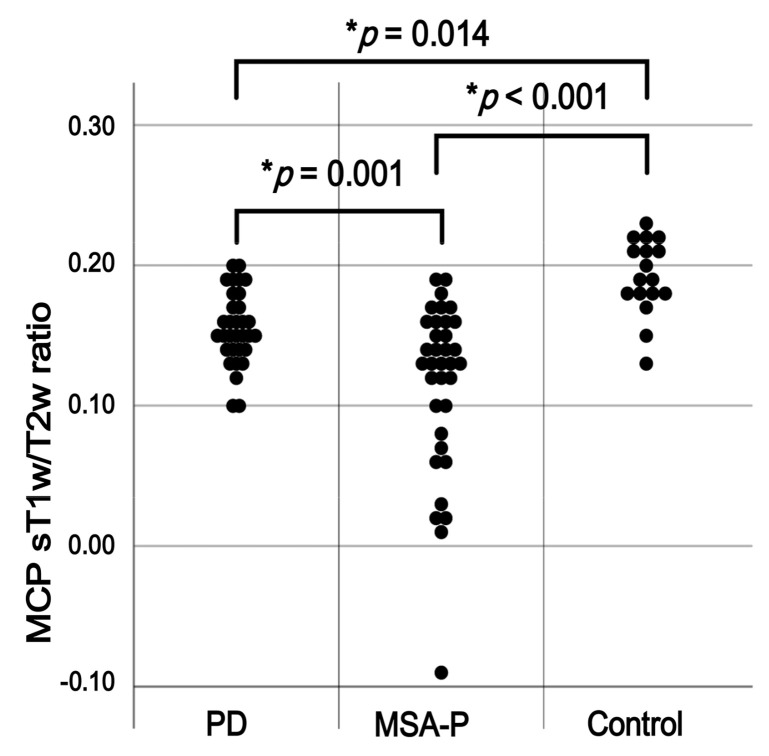
Comparison of the middle cerebellar peduncle (MCP) standardized T1-weighted/T2-weighted (sT1w/T2w) ratio between patients with the parkinsonian subtype of multiple system atrophy (MSA-P), those with Parkinson’s disease (PD), and the controls. * *p* < 0.05.

**Figure 3 diagnostics-14-00201-f003:**
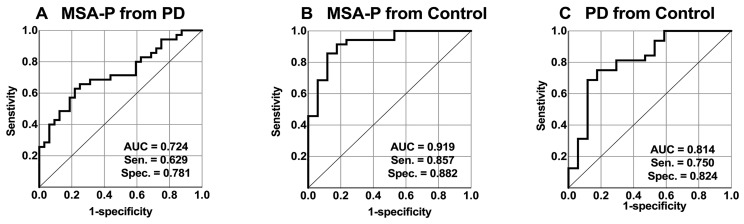
Middle cerebellar peduncle (MCP) standardized T1-weighted/T2-weighted (sT1w/T2w) ratio with receiver operating characteristic (ROC) curve analysis showing the area under the curve (AUC), sensitivity (Sen), and specificity (Spec). (**A**,**B**) ROC curves distinguishing the Parkinsonian subtype of multiple system atrophy (MSA-P) from Parkinson’s disease (PD) (**A**) and controls (**B**). (**C**) ROC curve differentiating PD from the controls.

**Figure 4 diagnostics-14-00201-f004:**
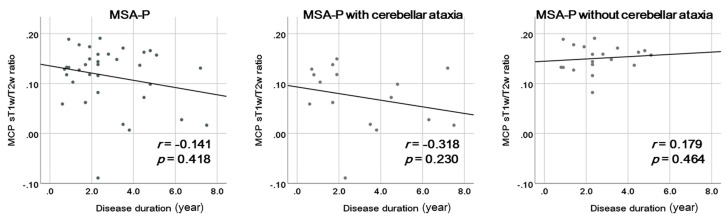
Correlations between the middle cerebellar peduncle (MCP) standardized T1-weighted/T2-weighted (sT1w/T2w) ratio and disease duration in patients with multiple system atrophy with parkinsonism (MSA-P) and the subgroups of patients with MSA-P with and without cerebellar ataxia.

**Figure 5 diagnostics-14-00201-f005:**
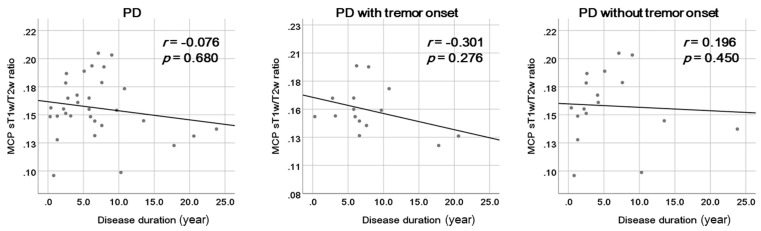
Correlations between the middle cerebellar peduncle (MCP) standardized T1-weighted/T2-weighted (sT1w/T2w) ratio and disease duration in patients with Parkinson’s disease (PD) and the subgroups of patients with PD with and without tremor onset.

**Table 1 diagnostics-14-00201-t001:** Demographic and clinical data of patients with Parkinson’s disease (PD), those with the Parkinsonian subtype of multiple system atrophy (MSA-P), and controls.

	Group (*n*)			*p*-Value for Group Comparisons	*p*-Value for Post Hoc Group Comparisons		
	PD (*n* = 32)	MSA-P (*n* = 35)	Control (*n* = 17)		PD vs. MSA-P	PD vs. Control	MSA-P vs. Control
Sex distribution (male/female) ^1^	12/20	13/22	9/8	0.503	0.976	0.299	0.279
Age at MRI (years, median, range) ^2^	68.5 (49–88)	68.0 (46–79)	63.0 (44–80)	0.321	0.806	0.115	0.173
Disease duration (years, median, range) ^3^	5.9 (0.3–23.8)	2.3 (0.7–7.5)	NA	0.001	NA	NA	NA
Age at onset (years, median, range) ^3^	61.0 (43–81)	65.0 (43–76)	NA	0.081	NA	NA	NA
Hoehn and Yahr stage (median, range) ^4^	3 (1–4)	3 (2–4)	NA	0.161	NA	NA	NA
LEDD (mg, median, range) ^3^	300.0 (0–2038.9)	0 (0–1472.0)	NA	0.007	NA	NA	NA

MRI, magnetic resonance imaging; NA, not applicable; LEDD, levodopa equivalent daily dose. ^1^ Chi-square test with post hoc tests adjusted for multiple comparisons (*p* < 0.05/3 = 0.0167). ^2^ Nonparametric tests (Kruskal–Wallis one-way ANOVA with post hoc Welch’s *t*-tests adjusted for multiple comparisons; *p* < 0.05/3 = 0.0167). ^3^ Mann–Whitney *U* test. ^4^ Chi-square test.

## Data Availability

Data are unavailable due to privacy concerns.

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
