# Peer review of "Differentiation between Parkinson’s Disease and the Parkinsonian Subtype of Multiple System Atrophy Using the Magnetic Resonance T1w/T2w Ratio in the Middle Cerebellar Peduncle"

_diagnostics, 2024, doi:10.3390/diagnostics14020201_

Round 1
Reviewer 1 Report
Comments and Suggestions for Authors
Wang et al. present an interesting study of using sT12w/T2w in MCP to evaluate MSA-P and PD. This study is well-written. The P-value in Table 1 is confusing. There are three groups for comparison in sex distribution and Age at MRI. The presented p-value should be define in both group.
Reviewer 2 Report
Comments and Suggestions for Authors
There are some important methodological issues, relating to: sample size, diagnosis and differences in disease duration among patient groups. Moreover, important data are missing, such as pharmacological treatments with levodopa or dopamine agonists (LEDD) and the severity of motor symptoms was not evaluated in PD and MSA patients. All these methodological issues make difficult to interpret results. I suggest the authors to address the following points
Sample size: I would like to receive an explanation on the strategies of sample size by the authors, because MSA prevalence is about 4 per 100,000 population, which is the prevalence of MSA-P? (2 patients per 100,000 individuals?) PD prevalence is also 1-2 per 1,000 individuals, but they manage to enroll 35 patients with MSA-P, 32 patients with PD and 17 controls (???)…These numbers seem to be unlikely.
Diagnosis of MSA-P and PD: How many patients with MSA-P show the cerebellar ataxia? Could the authors explain because they are sure that the diagnosis of MSA-P with cerebellar ataxia is correct? Could one patient with MSA and cerebellar ataxia be classified as MSA-C? How many patients with PD show the tremor at onset of disease?
Motor and non-motor symptoms: No assessment of motor symptoms was provided for patients with PD and MSA-P or staging of disease (e.g. Hoehn and Yahr scale). Please, the authors should provide a description of symptoms showed from patients with PD and MSA, as well as a motor evaluation.
Disease duration: The range of disease duration of both patient groups is very large… The authors should clarify and discuss the criteria used for inclusion and why they believe such ranges in terms of duration do not significantly affect results.
Inclusion/exclusion criteria: authors should clearly describe the inclusion and exclusion criteria for each group.
Comments on the Quality of English LanguageMinor editing of English language required
Reviewer 3 Report
Comments and Suggestions for Authors
Jiaqi Wang et al. investigated the distinction in the specific parameter (MR sT1w/T2w ratio) of the middle cerebellar peduncle between Parkinson’s disease (PD) and multiple system atrophy with parkinsonism (MSA-P). They discovered that the sT1w/T2w ratio holds diagnostic value not only in distinguishing MSA-P from the control group but also in discriminating between MSA-P and PD. As highlighted in the manuscript, discriminating between MSA-P and PD remains challenging in clinical settings, making the presented data valuable for clinicians and researchers.
As discussed in the limitations, the sample size was small, and the differences did not reach statistical significance. However, addressing this point is not necessary in this study. A recommended follow-up study in the future would strengthen their findings, as it is speculated that the ratio will decline more rapidly in MSA-P compared to PD when each regression line is compared.
Major Points:
-
Referring to the ROC curve in Fig.3, the optimal combination of sensitivity and specificity for “MSA-P from PD” is approximately 0.6 and 0.2, respectively. Clarifying the threshold with such sensitivity and specificity would be of interest to the reviewer.
-
The y-intercept of the regression line for “MSA-P with cerebellar ataxia” is around 0.10, indicating that the ratio is already small even in the preclinical disease stage. This suggests that the utility of the sT1w/T2w ratio relies on the degree of cerebellar involvement as a lesion at disease onset. This aligns with the authors' earlier findings. The reviewer suggests discussing this point.
Minor Points:
-
Presenting each sT1w/T2w ratio for every representative image in Fig.1 would provide informative reference points.
-
The x-axis in Fig.4 and 5 requires units. Please add “m” or “month” to specify the unit of measurement.
Round 2
Reviewer 2 Report
Comments and Suggestions for Authors
The authors addressed my concerns about this manuscript. I think that the quality of this work is sufficient to be published as well as Scientific Soundness
Comments on the Quality of English Language
Quality of English language is enough